# MULTI-DOMAIN SELF-SUPERVISED LEARNING

## ABSTRACT

Contrastive self-supervised learning has recently gained significant attention owing to its ability to learn improved feature representations without the use of label information. Current contrastive learning approaches, however, are only effective when trained on a particular dataset, limiting their utility in diverse multi-domain settings. In fact, training these methods on a combination of several domains often degrades the quality of learned representations compared to the models trained on a single domain. In this paper, we propose a Multi-Domain Self-Supervised Learning (MDSSL) approach that can effectively perform representation learning on multiple, diverse datasets. In MDSSL, we propose a three-level hierarchical loss for measuring the agreement between augmented views of a given sample, agreement between samples within a dataset and agreement between samples across datasets. We show that MDSSL when trained on a mixture of CIFAR-10, STL-10, SVHN and CIFAR-100 produces powerful representations, achieving up to a 25% increase in top-1 accuracy on a linear classifier compared to single-domain self-supervised encoders. Moreover, MDSSL encoders can generalize more effectively to unseen datasets compared to both single-domain and multi-domain baselines. MDSSL is also highly efficient in terms of the resource usage as it stores and trains a single model for multiple datasets leading up to 17% reduction in training time. Finally, for multi-domain datasets where domain labels are unknown, we propose a modified approach that alternates between clustering and MDSSL. Thus, for diverse multi-domain datasets (even without domain labels), MDSSL provides an efficient and generalizable self-supervised encoder without sacrificing the quality of representations in individual domains.

## 1 INTRODUCTION

Self-supervised contrastive training (Chen et al., 2020; He et al., 2020; Misra & van der Maaten, 2020; Caron et al., 2020b) has become a popular paradigm for unsupervised representation learning as it shows impressive results on linear classification tasks, almost matching the performance of a supervised model trained from scratch. However, we find that current self-supervised models are only effective when trained on a single-domain. This can hinder their deployment in large scale real-world settings where data almost always comes from multiple diverse domains. We illustrate this issue in Table 1, where we show that a popular self-supervised model, SimCLR (Chen et al., 2020), trained on CIFAR-10 (Krizhevsky et al., a) does not generalize to other domains at the test time. We observe that the top-1 accuracy of a linear classifier significantly drops on unseen datasets. This means that a different self-supervised model needs to be trained for every new dataset, which can add significant computational overheads given that training these models often require large batch sizes and a large number of training epochs (Chen et al., 2020; He et al., 2020; Wu et al., 2018).

One potential solution for self-supervised learning on multi-domain datasets is to train the models on the *union* of all input domains. Unfortunately, this solution performs poorly and fails to obtain a good performance on every individual dataset and does not generalize well to unseen domains. To illustrate this, we trained SimCLR on the union of multiple datasets including CIFAR-10 (Krizhevsky et al., a), CIFAR-100 (Krizhevsky et al., b), SVHN (Netzer et al., 2011) and STL-10 (Coates et al.). The trained model is unfavorable as it significantly decreases the top-1 accuracy in all training datasets compared to the single-domain baselines (see Table 1).

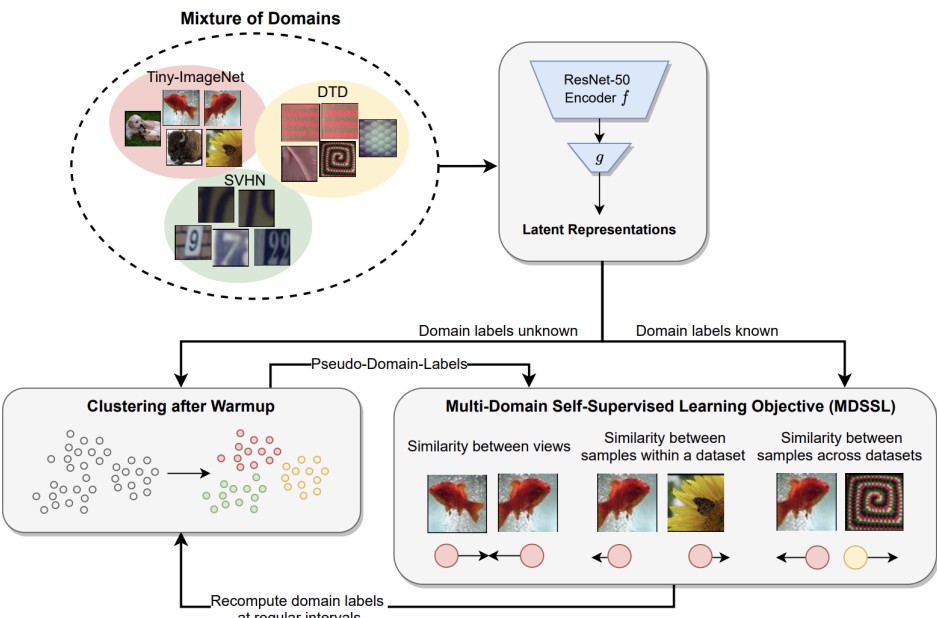

Figure 1: **Framework of Multi-Domain Self-Supervised Learning:** Let us consider our input as a mixture of domains, containing datasets from various sources. We introduce MDSSL, a three-level hierarchical self-supervised learning approach to perform representation learning on all of these domains at the same time. Using a standard ResNet-50 encoder, we learn latent representations that are optimized using the MDSSL objective. We support MDSSL under two setups - *with* and *without* domain labels. When domain labels are not available, we first cluster representations to identify pseudo-domain-labels and then train MDSSL. We use a *robust* clustering approach, while recomputing clusters at regular intervals.

To tackle these issues, we propose **Multi-Domain Self-Supervised Learning (MDSSL)**, a technique for obtaining a unified embedder that can be trained on multiple domains. In MDSSL, we train the model over the union of multiple datasets using a three-level hierarchical loss involving:

- **Embedding similarities of two views of a sample:** In the first level, we maximize agreement (i.e. the cosine similarity between $l_2$-normalized vectors) between embeddings of two augmented views of a given sample.

- **Embedding similarities of samples from a given dataset:** In the second level, we minimize the pairwise agreements between embeddings of all samples within a dataset.

- **Embedding similarities of samples from different datasets:** In the third level, we minimize the pairwise agreement between samples across all training datasets.

The first two levels ensure that the model learns high quality representations for each individual domain. The third level of the MDSSL loss encourages the model to learn distinguishable representations between domains. This approach assumes that domain labels are known during training.

We also extend MDSSL to more realistic multi-domain setups where domain labels are *unknown*. In such scenarios, we present an iterative approach that alternates between clustering and MDSSL at fixed intervals. We use clustering to detect pseudo-domain-labels for each training dataset and use these labels in the MDSSL loss. We also propose a robust version of clustering by reducing outlier noise which further improves the performance of MDSSL in an entirely unsupervised setup.

In summary, the goal of MDSSL is to compute improved latent representations of samples from multiple diverse datasets using a *single* self-supervised model (See Figure 1). We summarize our contributions as follows:

- We show that current self-supervised learning techniques such as SimCLR, under multi-domain setups, show degraded performance on downstream linear classification tasks and do not generalize well to unseen domains.

- We propose *Multi-Domain Self-Supervised Learning (MDSSL)* that uses a new loss function for self-supervised learning that supports training over multiple domains at once and pushes the model to learn distinguishable representations across datasets.

- We show that MDSSL trained on a mixture of CIFAR-10, STL-10, SVHN and CIFAR-100, shows a $25\%$ increase in top-1 accuracy and is more efficient (See Table 1).

- We also experiment over DTD and Tiny-ImageNet and show that MDSSL generalizes better to unseen domains of varying diversity compared to both single-domain SimCLR and multi-domain SimCLR.

- We propose an iterative approach combining MDSSL with clustering to train over multi-domain datasets without the use of domain labels.

- We further improve our clustering approach by introducing *robust* clustering that prevents outlier noise from affecting domain labels.

Table 1: Comparing SimCLR and MDSSL on single and multi-domain setups

| | Train Dataset | Top-1 Accuracy | | | | |
| | | CIFAR-10 | STL-10 | SVHN | CIFAR-100 | Average |
|---|---|---|---|---|---|---|
| | *Single-Domain Training* | | | | | |
| SimCLR | CIFAR-10 | **92.35** | 56.71 | 55.97 | 75.37 | 70.10 |
| | STL-10 | 71.05 | **77.58** | 46.06 | 63.81 | 64.62 |
| | SVHN | 62.83 | 46.77 | **92.42** | 48.27 | 62.57 |
| | CIFAR-100 | 79.58 | 55.27 | 61.16 | **90.29** | 71.57 |
| | *Multi-Domain Training* | | | | | |
| SimCLR | CIFAR-10, CIFAR-100, SVHN, STL-10 | 82.30 | 61.41 | 66.65 | 73.41 | 70.94 |
| MDSSL | CIFAR-10, CIFAR-100, SVHN, STL-10 $(\lambda_1 = 1, \lambda_2 = 0.1)$ | **88.45** | **65.95** | **75.35** | **83.05** | **78.20** |

## 2 RELATED WORK

Supervised classification techniques involve minimizing a loss function (e.g. the cross-entropy loss) to match model predictions to true labels. Unsupervised classification methods, on the other hand, learn to classify data without the use of training labels, usually with the use of clustering techniques (Bojanowski & Joulin, 2017; Dosovitskiy et al., 2014; YM. et al., 2020; Bautista et al., 2016; Caron et al., 2018; 2019; Huang et al., 2019).

More recently, new unsupervised techniques called self-supervised representation learning have been proposed. A self-supervised model learns by observing every instance of the given data and assigns its own labels, and then performs a classification task (Bojanowski & Joulin, 2017; Dosovitskiy et al., 2014; Wu et al., 2018; Dosovitskiy et al., 2016). To simplify the complexity of instance-level classification, a memory bank (Wu et al., 2018; He et al., 2020) can be used with the help of contrastive learning (Gutmann & Hyvärinen, 2010; Hjelm et al., 2019; van den Oord et al., 2019; Grill et al., 2020). Contrastive learning (Arora et al., 2019; Tosh et al., 2021; Bachman et al., 2019) is a temperature-controlled cross-entropy loss between positive pairs of *similar* samples and negative pairs of *dissimilar* samples. Positive pairs are usually considered as multiple transformations (views) (Tian et al., 2020) of a given sample using stochastic data augmentation. SimCLR (Chen et al., 2020) shows that contrastive learning can be done without the use of a memory bank, using the samples within a batch, if we have large enough batches. SwAV (Caron et al., 2020a) uses a

mixture of contrastive learning and clustering to form a swapped prediction problem that can learn even with very small batch sizes. Finally, contrastive learning can benefit from training labels, if available with a simple modification of contrasting between samples within a class and taking samples of other classes as negatives (Khosla et al., 2020). Each of these approaches show remarkable linear classification accuracy on single-domain setups.

Extending self-supervised learning to multiple diverse domains, other than ImageNet (Russakovsky et al., 2015), is a relatively less explored topic (Wallace & Hariharan, 2020). When multiple related domains are available during training, a possible approach is to use mutual information to simultaneously encode common invariant information and domain-specific information of each image (Feng et al., 2019). In our paper, we focus on a general setup where we combine diverse unrelated domains and evaluate individual domain-specific tasks.

## 3 MULTI-DOMAIN SELF-SUPERVISED LEARNING WITH DOMAIN LABELS

In this section, we define the Multi-Domain Self-Supervised Learning (MDSSL) paradigm for $D$ training datasets where domain labels are known. We define $\mathbf{x}_i^d \in \mathbb{R}^r$ and $\tilde{\mathbf{x}}_i^d \in \mathbb{R}^r$ as two transformed views of the $i^{th}$ sample from the $d^{th}$ dataset, $d \in \{1, ..., D\}$. Similar to SimCLR, we use a base encoder $f(.)$ and a two-layer MLP projection head $g(.)$ to map a given sample into the latent space. We define the latent representations of the two views of the $i^{th}$ sample from the $d^{th}$ dataset as $\mathbf{z}_i^d = f(g(\mathbf{x}_i^d)) \in \mathbb{R}^{r'}$ and $\tilde{\mathbf{z}}_i^d = f(g(\tilde{\mathbf{x}}_i^d)) \in \mathbb{R}^{r'}$ where $r'$ is the size of each latent representation. We represent mini-batches containing $2N$ samples (2 views per sample) from $D$ datasets as a matrix $\mathbf{X} \in \mathbb{R}^{2ND \times r}$, whose corresponding latent representation is denoted by $\mathbf{Z} \in \mathbb{R}^{2ND \times r'}$.

We then calculate a similarity matrix $\mathbf{S} \in \mathbb{R}^{(2ND) \times (2ND)}$ that contains the exponential cosine similarity scaled by a temperature parameter $\tau$, between all the latent representations in a given batch. The $(i, j)^{th}$ element of $\mathbf{S}$ is:

$$\mathbf{S}_{(i,j)} := \exp\left(\frac{1}{\tau} \frac{\mathbf{z}_i^T \mathbf{z}_j}{\|\mathbf{z}_i\|\|\mathbf{z}_j\|}\right) \tag{1}$$

where $\mathbf{z}_i \in \mathbb{R}^{r'}$ and $\mathbf{z}_j \in \mathbb{R}^{r'}$ are the $i^{th}$ and $j^{th}$ row of $\mathbf{Z}$, respectively.

$S_i^d$ represents the cosine similarity between $\mathbf{z}_i^d$ and $\tilde{\mathbf{z}}_i^d$. $S_{ij}^{d,d'}$ represents the cross-dataset cosine similarity between $\mathbf{z}_i^d$ and $\mathbf{z}_j^{d'}$ where $d, d' \in \{1, ..., D\}$. MDSSL aims to solve the following optimization problem:

$$\max_\theta \left(\frac{1}{ND} \sum_{d=1}^{D} \sum_{i=1}^{N} \log S_i^d\right) \tag{2}$$

$$- \lambda_1 \left(\frac{1}{2ND} \sum_{d=1}^{D} \sum_{i=1}^{2N} \log \sum_{j=1}^{2N} \mathbb{1}_{j \neq i} S_{ij}^{d,d}\right) \tag{3}$$

$$- \lambda_2 \left(\frac{1}{2ND} \sum_{d=1}^{D} \sum_{i=1}^{2N} \log \sum_{d'=1}^{D} \sum_{j=1}^{2N} \mathbb{1}_{d' \neq d} S_{ij}^{d,d'}\right) \tag{4}$$

where $\theta$ is the set of model parameters and $0 < \lambda_1 \leq 1$ and $\lambda_2 \geq 0$ are tunable regularization parameters. This is a three-level hierarchical loss. (2) maximizes the similarity between two transformed views ($\mathbf{x}_i^d$ and $\tilde{\mathbf{x}}_i^d$) in the latent space. (3) minimizes the similarity between every pair of samples ($\mathbf{x}_i^d$ and $\mathbf{x}_j^d$) within a dataset. (4) minimizes the similarity between pairs of samples ($\mathbf{x}_i^d$ and $\mathbf{x}_j^{d'}$) across different datasets. When $\lambda_1 = 1$ and $\lambda_2 = 0$, this optimization is simplified to SimCLR (Chen et al., 2020). Therefore, we use $\lambda_1 = 1$ and $\lambda_2 = 0$ as the baseline in all our experiments.

$\lambda_1$ helps us control the extent to which we want to minimize the similarity within a domain. We empirically observe that relaxing $\lambda_1$ from the SimCLR baseline ($\lambda_1 = 0$) to a value slightly less than 1, in fact generates better structure in the latent space by clustering samples within a domain relatively closer compared to samples outside a domain (See Appendix Section A.3). $\lambda_2$ should always be non-negative as we always want to minimize the agreement between samples of different

datasets. We can implement the MDSSL loss efficiently as it only involves calculating the similarity matrix $\mathbf{S}$ once and then selecting elements according to each term mask. As the number of datasets increases, the size of $\mathbf{S}$ increases, gradually increasing the running time of MDSSL.

## 3.1 Experimental Setup

We use ResNet-50 (He et al., 2016) as the base encoder ($f(.)$) and a 2-layer MLP projection head ($g(.)$) for all of our experiments. For data augmentation, we use a combination of *random crop*, *random horizontal flip*, *random color distortion* and *random Gaussian blur*. In all experiments, the latent representations are in a 128-dimensional space and $\tau = 0.1$. We optimize our loss using LARS optimizer (You et al., 2017) with a learning rate of 4 and weight decay of $10^{-6}$. We train with a batch size of 1024 and train over 48,000 iterations. We experiment with the following datasets: CIFAR-10 (Krizhevsky et al., a), CIFAR-100 (Krizhevsky et al., b), STL-10 (Coates et al.), SVHN (Netzer et al., 2011), Tiny-ImageNet (Le & Yang, 2015) and DTD (Describable Textures) (Cimpoi et al., 2014). We resize all images to 32x32 in all our experiments. We use Nvidia GeForce RTX 2080 GPUs. We measure the quality of representations using the linear evaluation protocol (Kolesnikov et al., 2019; Bachman et al., 2019; van den Oord et al., 2019) where we train a linear classifier on top of frozen MDSSL representations and compute the top-1 accuracy of each domain-specific classification task. Since we train over multiple domains, we compute the top-1 accuracy over each domain to evaluate the overall model performance.

Table 2: Resource utilization of SimCLR and MDSSL when trained on CIFAR-10, STL-10, SVHN and CIFAR-100

| Resource | Single-Domain SimCLR | MDSSL |
|---|---|---|
| Training Time (hours) | 42.41 | **34.95** (-17.59%) |
| Disk Memory (MB) | 968 | **242** (-75%) |
| Compute (GPUs) | 4 | **2** (-50%) |

## 3.2 MDSSL Performance compared to SimCLR baseline

In this section, we analyze the performance of MDSSL and compare it to SimCLR trained on single domains (referred to as the single-domain SimCLR) and multiple domains (referred to as the multi-domain SimCLR). Table 1 summarizes our results on CIFAR-10, STL-10, SVHN and CIFAR-100. We observe that, single-domain SimCLR models generalize poorly on unseen datasets. For example, SimCLR trained on CIFAR-10 achieves 92.35% top-1 accuracy on CIFAR-10 samples but only 55.97% on SVHN samples.

We also observe that the multi-domain SimCLR model shows a degraded performance when evaluated on each individual training domain. For example, SimCLR trained on the union of samples from CIFAR-10, STL-10, SVHN and CIFAR-100 achieves 82.30% top-1 accuracy on CIFAR-10, significantly lower than the performance of the single-domain SimCLR model trained on CIFAR-10. Our method, MDSSL, shows a significant improvement compared to the multi-domain SimCLR and almost matches the baseline accuracy of single-domain SimCLR models on some of the training domains. Among the average top-1 accuracy, we observe up to 25% improvement from the single-domain SimCLR and a 10% improvement from the multi-domain SimCLR (See Table 1). SimCLR would require us to train 4 different single-domain models for these datasets and therefore requires more compute, memory and time. MDSSL, being a unified model, significantly outperforms SimCLR in terms of resource utilization as shown in Table 2. This makes MDSSL an efficient solution in limited resource environments.

## 3.3 Generalization to Unseen Datasets

In this section we evaluate the generalization capacity of MDSSL to unseen domains. We consider two setups: in the first case, we use CIFAR-10, STL-10 and SVHN as training datasets (domains containing less diverse datasets as their number of classes are $\leq 10$) and evaluate the model performance on unseen datasets of CIFAR-100, DTD and Tiny ImageNet (highly diverse datasets whose number of classes are $> 10$). In the second case, we use CIFAR-100, DTD and Tiny ImageNet as

Table 3: Generalization of SimCLR and MDSSL to unseen domains

| | Train Dataset | Top-1 Accuracy | | | | | | Average | |
| --- | --- | --- | --- | --- | --- | --- | --- | --- | --- |
| | | CIFAR-10 | STL-10 | SVHN | CIFAR-100 | DTD | Tiny-ImageNet | | |
| *SimCLR* | CIFAR-10 | **92.35** | 56.71 | 55.97 | 75.37 | 40.50 | 19.95 | 56.80 | |
| | STL-10 | 71.05 | **77.58** | 46.06 | 63.81 | 39.22 | 21.41 | 53.18 | |
| | SVHN | 62.83 | 46.77 | **92.42** | 48.27 | 36.47 | 13.42 | 50.03 | |
| | CIFAR-100 | 79.58 | 55.27 | 61.16 | **90.29** | 42.24 | 21.36 | 58.31 | |
| | DTD | 64.95 | 51.68 | 49.86 | 55.98 | **50.43** | 19.40 | 48.71 | |
| | Tiny-ImageNet | 81.67 | 63.20 | 53.75 | 82.69 | 44.03 | **37.99** | **60.55** | |
| | ImageNet (250K) | 68.16 | 75.43 | 49.09 | 50.03 | 50.57 | 21.00 | 52.38 | |
| | *Multi-Domain Training* | | | | | | | Average (Training domains) | Average (Unseen domains) |
| *SimCLR* | CIFAR-10, STL-10, SVHN | 83.96 | 63.23 | 72.10 | 71.72 | 47.94 | 22.67 | 73.09 | 47.44 |
| *MDSSL* | CIFAR-10, STL-10, SVHN ($\lambda_1 = 0.9, \lambda_2 = 0.1$) | **87.50** | **65.58** | **88.05** | **76.48** | **49.36** | **24.49** | **80.37** | **50.11** |
| *SimCLR* | CIFAR-100, DTD, Tiny ImageNet | 77.27 | 59.22 | 68.06 | 75.72 | 51.82 | 28.40 | 51.98 | 68.18 |
| *MDSSL* | CIFAR-100, DTD, Tiny ImageNet ($\lambda_1 = 0.9, \lambda_2 = 0.05$) | **81.92** | **62.89** | **72.35** | **83.93** | **54.77** | **30.18** | **56.29** | **72.38** |
| *SimCLR* | ImageNet (250K), CIFAR-100, SVHN | 76.95 | 74.87 | 59.82 | 69.95 | 52.11 | 27.99 | 64.88 | 57.98 |
| *MDSSL* | ImageNet (250K), CIFAR-100, SVHN ($\lambda_1 = 0.9, \lambda_2 = 0.1$) | **81.86** | **77.01** | **78.64** | **80.16** | **55.30** | **33.04** | **79.40** | **61.80** |

our training datasets and assess the performances on CIFAR-10, STL-10 and SVHN. We also add results on ImageNet (250K) which contains 1000 classes, each including 250 samples resized to 32x32.

Table 3 summarizes our results. Among the single-domain SimCLR models, we observe that Sim-CLR trained on Tiny-ImageNet generalizes relatively better than other single-domain models since Tiny-ImageNet is comparatively larger and most diverse. However, the drop in top-1 accuracy of unseen domains from the baseline is very significant even for the single-domain SimCLR trained on Tiny-ImageNet (42% drop for SVHN). Similarly, ImageNet (250K) also poorly generalizes to unseen domains.

In our first multi-domain setup (with training datasets of CIFAR-10, STL-10 and SVHN), we observe that although these training datasets are relatively less diverse, MDSSL generalizes remarkably well on more diverse datasets like CIFAR-100, DTD and Tiny-ImageNet. MDSSL also outperforms the multi-domain SimCLR in all domains (training and unseen). We observe a similar improvement when we train MDSSL on CIFAR-100, DTD and Tiny-ImageNet and on ImageNet (250K), CIFAR-100 and SVHN. MDSSL outperforms both single and multi-domain SimCLR in terms of generalization capacity. These results highlight that MDSSL is a favorable solution that achieves good accuracy on training domains and generalizes well to unseen domains.

## 3.4 EFFECT OF NUMBER OF TRAINING DATASETS

In this section, we discuss the behavior of MDSSL as we increase the number of training domains. We train MDSSL on CIFAR-10 and CIFAR-100 (2-domain baseline). We then add STL-10, SVHN, Tiny-ImageNet and DTD datasets one by one and train MDSSL.

In Figure 2, we observe that as the number of datasets increases, the top-1 accuracy also increases and finally beats the single-domain baseline. Therefore, MDSSL benefits from training over a large number of datasets.

### 3.5 Hyperparameter Selection

The MDSSL loss is controlled by two regularizers $\lambda_1$ and $\lambda_2$, as shown in Section 3. When $\lambda_1 = 1$ and $\lambda_2 = 0$, MDSSL boils down to our baseline, SimCLR (Chen et al., 2020). As we decrease $\lambda_1$ while fixing $\lambda_2 = 0$, we observe that the in-distribution similarity increases (See Appendix Section A.3) and eventually, all samples show a mutual similarity of $1$. Consequently, the top-1 accuracy quickly degrades from the baseline as shown in the first plot in Figure 3. This behavior can be explained by Term 3 of the MDSSL loss in Section 3 which measures the mutual similarity between all samples within a dataset. Therefore, we fix $\lambda_1 \geq 0.9$ so that it mildly increases in-distribution similarity without significantly affecting the top-1 accuracy. We utilize Term 4 of the MDSSL loss by controlling $\lambda_2$, to ensure that domains are more distinguishable in the latent space. In Figure 3, the second plot shows the top-1 accuracy as we increase $\lambda_2$. The top-1 accuracy rises steadily at first, and then drops at around $\lambda_2 = 0.2$. This is because after a certain threshold, the in-distribution representations become too similar which makes them harder to classify. Therefore, a good balance should be found between $\lambda_1$ and $\lambda_2$ such that we achieve favourable top-1 accuracy.

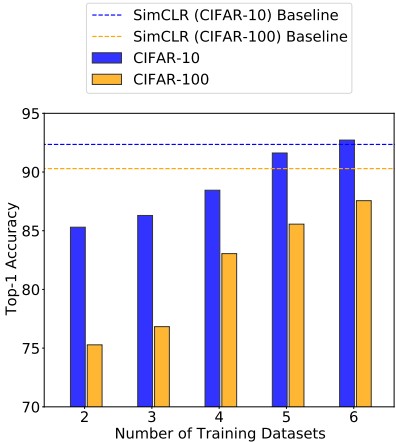

Figure 2: **Effect of number of training datasets.** In this plot, we show that when the number of training datasets increases in MDSSL, the top-1 accuracy increases and eventually beats the SimCLR single-domain baseline, marked by dotted lines. We train MDSSL on CIFAR-10, CIFAR-100, STL-10, SVHN, Tiny-ImageNet and DTD.

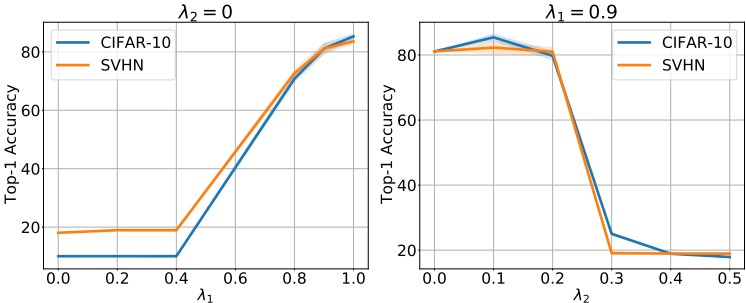

Figure 3: **Top-1 accuracy of MDSSL trained on CIFAR-10 and SVHN with varying $\lambda_1$ (similarity within dataset) and $\lambda_2$ (similarity across datasets).** In the first plot, we observe that increasing $\lambda_1$ from $-1$ (SimCLR Chen et al. (2020)) quickly drops the top-1 accuracy since samples within a dataset become more and more indistinguishable. In the second plot, when $\lambda_1 = -0.9$, the top-1 accuracy steadily improves with $\lambda_2$ until a threshold ($\lambda_2 = 0.2$) and then drops. These plots show that there is a sweet-spot in selecting $\lambda_1$ and $\lambda_2$ such that we achieve high top-1 accuracy.

## 4 Multi-Domain Self-Supervised Learning without Domain Labels

Most real-world multi-domain datasets are unlabelled (i.e., domain label information is not available). In this section, we develop an extension of MDSSL for such setups by identifying pseudo domain labels via a clustering approach in the latent space. As it is common in clustering, we assume the number of domains (denoted by $M$) is known.

In the MDSSL loss (especially in Term 4), we need domain labels to compute pairwise similarities of samples from two different domains. To achieve this, we first treat the problem as single-domain self-supervised learning and warm up the MDSSL encoder for the first few training iterations using the optimization described in Section 3 with $=\lambda_1 = 1$, $\lambda_2 = 0$ and $D = 1$ (i.e., SimCLR training on one domain). This warm up helps us get somewhat distinguishable representations for samples between $M$ domains and the number of iterations to warm up is determined empirically. At the end of the warm up, we cluster the latent representations of the entire multi-domain dataset into $M$ clusters using K-Means clustering (Hartigan & Wong, 1979). Using these clusters as pseudo-domain-labels, we continue training the encoder under the MDSSL loss with $\lambda_1 \leq 1$, $\lambda_2 > 0$ and $D = M$. As the training progresses, MDSSL improves the latent structure and therefore, we recompute clusters multiple times (determined empirically) as the training progresses to ensure that improved domain labels are used.

In practice, we observe that clustering does not provide $100\%$ accurate domain labels, especially for datasets that are distributionally similar such as CIFAR-10 and STL-10. In such cases, we propose to use a *robust* clustering approach coupled with MDSSL to prevent outlier clustering noise from affecting the MDSSL training. Let us consider a MDSSL encoder that is warmed up on a multi-domain dataset containing $M$ domains. We cluster the representations of this dataset into $M$ clusters with centroids $\mathbf{c}_1, \mathbf{c}_2, \ldots, \mathbf{c}_M$. Before assigning pseudo-domain-labels to each representation, we first determine if they are outliers or not. If so, we ignore these samples in training MDSSL in the next round. We say a latent sample $\mathbf{z}_i$ is *not* an outlier if it is significantly closer to one of the clustering centroids compared to another. Concretely, $\mathbf{z}_i$ is not an outlier if

$$\max \left\{ \frac{\|\mathbf{z}_i - \mathbf{c}_m\|^2}{\|\mathbf{z}_i - \mathbf{c}_n\|^2} : 1 \leq m \leq M, 1 \leq n \leq M \right\} > 1 + \epsilon \tag{5}$$

where $\epsilon \geq 0$ is defined as an outlier threshold. When $\epsilon$ is high, it means that the given sample is close to its respective centroid. When $\epsilon$ approaches $0$, it indicates that the sample is almost equidistant from at least two centroids and therefore, may not be reliably clustered into one. We ignore such samples in MDSSL training. When we perform clustering for the first time, we start with $\epsilon = 1$ and each time we repeat clustering, we decay its value exponentially such that it approaches $0$ by the end of training to ensure that at the end, all samples contribute to the MDSSL training.

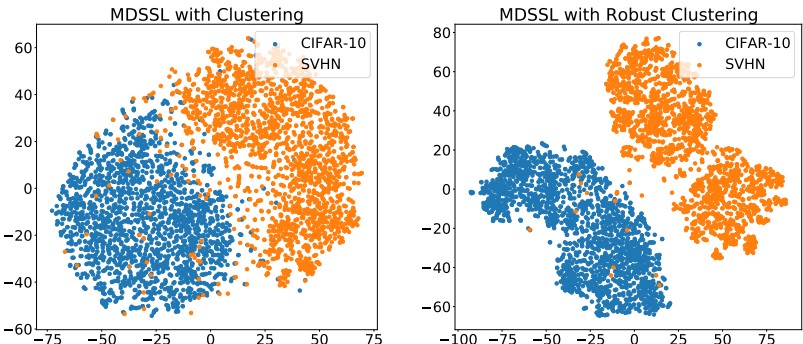

Figure 4: **Latent Space of MDSSL with Clustering:** We use TSNE to visualize the latent space of MDSSL with clustering on CIFAR-10 and SVHN. We observe that clustering helps us distinguish between domains and this improves when we apply *robust* clustering as shown above.

## 4.1 PERFORMANCE OF MDSSL TRAINED WITHOUT DOMAIN LABELS

Our experimental setup for training MDSSL without domain labels remains the same as the one we explain in Section 3.1. We perform the SimCLR warmup for $480$ iterations and update the clusters every $2,400$ iterations going forward. In this section, we consider two mixtures for training datasets: (i) CIFAR-10 and SVHN (containing visually dissimilar samples), (ii) CIFAR-10, STL-10

(containing visually similar samples). In Figure 4, we plot the TSNE of CIFAR-10 and SVHN sample embeddings while using either clustering or robust clustering approaches. We observe that with clustering, we achieve a reasonable separation between domains although there are several outliers. These outliers are significantly reduced while using robust clustering as described in Equation 5. As a result, the clusters are quite well defined and easily distinguishable.

In Figure 5, we plot the top-1 accuracy of 5 training setups: single-domain SimCLR, multi-domain SimCLR, MDSSL (with domain labels), MDSSL (with clustering) and MDSSL (with robust clustering). When trained on CIFAR-10 and SVHN, we observe that MDSSL with clustering outperforms SimCLR on both datasets and on CIFAR-100 which is an unseen domain. We also observe that MDSSL with clustering seems to generalize better to CIFAR-100 compared to MDSSL. MDSSL with clustering also outperforms SimCLR when trained on CIFAR-10 and STL-10 which are more visually similar. We also observe that applying robust clustering shows an improvement on all domains including unseen domains (CIFAR-100). These results highlight that clustering is a useful approach to identify pseudo-domain-labels and when coupled with MDSSL, it helps us learn better representations for seen and unseen domains.

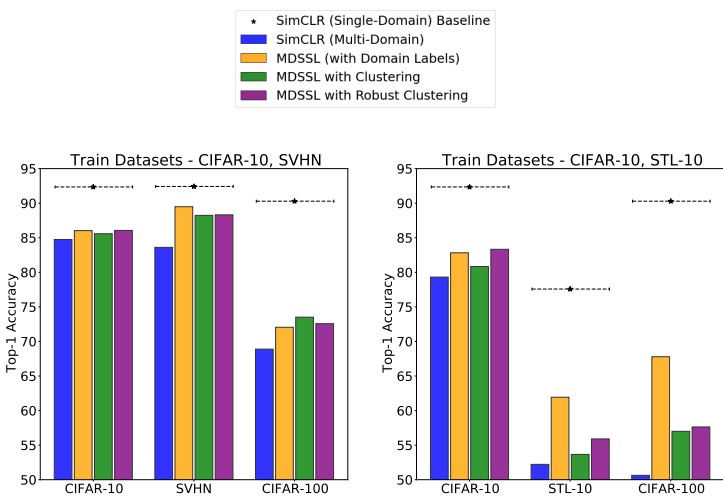

Figure 5: **MDSSL trained with Clustering:** We train MDSSL using clustering and *robust* clustering on CIFAR-10 and SVHN (left) and CIFAR-10 and STL-10 (right). We observe that, although domain labels are not used in clustering, we are able to improve the performance of MDSSL compared to SimCLR.

## 5  DISCUSSION

We propose Multi-Domain Self-Supervised Learning (MDSSL), as a unifying approach to compute self-supervised representations for a range of datasets. We support training MDSSL under two setups: with domain labels and without domain labels. We show that MDSSL achieves up to a $25\%$ increase in top-1 accuracy with linear evaluation compared to the SimCLR baseline on a combination of CIFAR-10, STL-10, SVHN and CIFAR-100. We also show that MDSSL is significantly more efficient than SimCLR in terms of resource (time, compute and memory) utilization, generalizes better than SimCLR in multi-domain setting, and benefits from an increase in the number of training datasets. In addition, we propose two versions of clustering that can be coupled with MDSSL when training over multiple domains without the use of domain labels. MDSSL achieves good performance even under these entirely unsupervised setups. Our unified approach, MDSSL, is general-purpose, enables training on diverse multi-domain settings, and can obtain meaningful embeddings achieving state-of-the-art results both on seen (training) and unseen benchmark datasets.

## 6 REPRODUCIBILITY STATEMENT

We share our code in the supplementary materials. We also provide several implementation details to ensure reproducibility of all our experiments. In Sections 3.1 and A.5, we provide a detailed explanation of our training setup including the architecture of our encoder, optimizers, learning rate schedule and training hyperparameters. We explain the process of hyperparameter selection in Sections 3.5 and A.3.

## 7 ETHICS STATEMENT

We use only publicly available datasets which involve classification tasks on general objects, vehicles, animals, etc. To the best of our knowledge, our work does not have a negative impact on our society or any societal group. However, as with all machine learning models, MDSSL should not be used on datasets that are inherently biased or involve harmful tasks that target or affect any particular regional, cultural or societal group. Therefore, before running MDSSL, one must select datasets and downstream tasks such that they are safe and do not amplify any social biases.

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

# A  APPENDIX

## A.1  EXPERIMENTS ON NON-OBJECT-FOCUSED DATASETS

In this section we discuss the results on 3 datasets that are not object-focused i.e, EuroSAT (Helber et al., 2019), Chest X-Ray (Pneumonia) (Kermany et al., 2018) and DTD (Cimpoi et al., 2014). These datasets are understandably not generalizable to unseen domains as shown in Table 4. However, under a multi-domain setup where we combine all of these domains, MDSSL shows a modest improvement compared to multi-domain SimCLR and almost matches the single-domain baselines.

Table 4: Comparing SimCLR and MDSSL on diverse non-object-focused datasets

|  | Train Dataset | Top-1 Accuracy | | | |
|---|---|---|---|---|---|
|  |  | EuroSAT | ChestXRay | DTD | Average |
|  | *Single-Domain Training* | | | | |
|  | EuroSAT | **88.95** | 93.57 | 45.75 | 76.09 |
|  | ChestXRay | 85.03 | **95.29** | 46.20 | 75.50 |
|  | DTD | 86.11 | 93.41 | **50.43** | 76.65 |
|  | *Multi-Domain Training* | | | | |
| *SimCLR* | EuroSAT, ChestXRay, DTD | 86.02 | 94.27 | 46.78 | 75.69 |
| *MDSSL* | EuroSAT, ChestXRay, DTD ($\lambda_1 = 0.9, \lambda_2 = 0.15$) | **87.10** | **94.43** | **50.23** | **77.25** |

## A.2  COMPARING MDSSL WITH SIMCLR PRE-TRAINED ON IMAGENET

In this section, we use a pre-trained SimCLR encoder from Pytorch Lightning Bolts (Falcon et al., 2019) and train a linear classifier on several unseen datasets. We resize all images to 32x32 during linear classification to maintain consistency with the rest of our experiments. We realize this may be an unfair comparison since the encoder is pre-trained on 224x224 images. Nevertheless, we observe a significant improvement in generalization of MDSSL over SimCLR pre-trained on full-sized ImageNet on all datasets (See Table 5).

Table 5: Comparing SimCLR pre-trained on full ImageNet with MDSSL

|  | Train Dataset | Top-1 Accuracy | | | | | | |
|---|---|---|---|---|---|---|---|---|
|  |  | CIFAR-10 | STL-10 | SVHN | CIFAR-100 | DTD | Tiny-ImageNet | Average |
|  | ImageNet | 68.21 | 58.72 | 49.05 | 50.11 | 47.36 | 20.85 | 49.05 |
|  | *Multi-Domain Training* | | | | | | | |
| *MDSSL* | CIFAR-10, STL-10, SVHN ($\lambda_1 = 0.9, \lambda_2 = 0.1$) | **87.50** | **65.58** | **88.05** | 76.48 | 49.36 | 24.49 | **65.24** |
| *MDSSL* | CIFAR-100, DTD, Tiny ImageNet ($\lambda_1 = 0.9, \lambda_2 = 0.05$) | 81.92 | 62.89 | 72.35 | **83.93** | **54.77** | **30.18** | 64.34 |

## A.3 HYPERPARAMETER SELECTION

In Figure A.1, we plot the similarity matrices of class-averaged samples of MDSSL trained on CIFAR-10 and SVHN by fixing $\lambda_2 = 0$ and varying $\lambda_1$. As explained in Section 3, higher $\lambda_1$ increases the similarity of samples within a domain. We observe that when $\lambda_1 = -1$ and $\lambda_2 = 0$ (SimCLR), the similarity within a domain is comparable with the similarity across domains, meaning that, domains are indistinguishable. As we increase $\lambda_1$, we see that the similarity within domains increases and eventually, all samples show a mutual similarity of 1.

In Figure A.1, although similarity within domains increases, we still cannot distinguish between domains. To achieve this, we utilize Term 4 of the MDSSL loss by controlling $\lambda_2$. In Figure A.2, we vary $\lambda_2$ and fix $\lambda_1 = -1$ (first row) and $\lambda_1 = -0.9$ (second row). As $\lambda_2$ increases, the similarity across domains decreases and each domain become clearly distinguishable. When $\lambda_1 = -0.9$, the effect is seen even at lower values of $\lambda_2$, as MDSSL learns to simultaneously increase similarity within domains while decreasing similarity across domains.

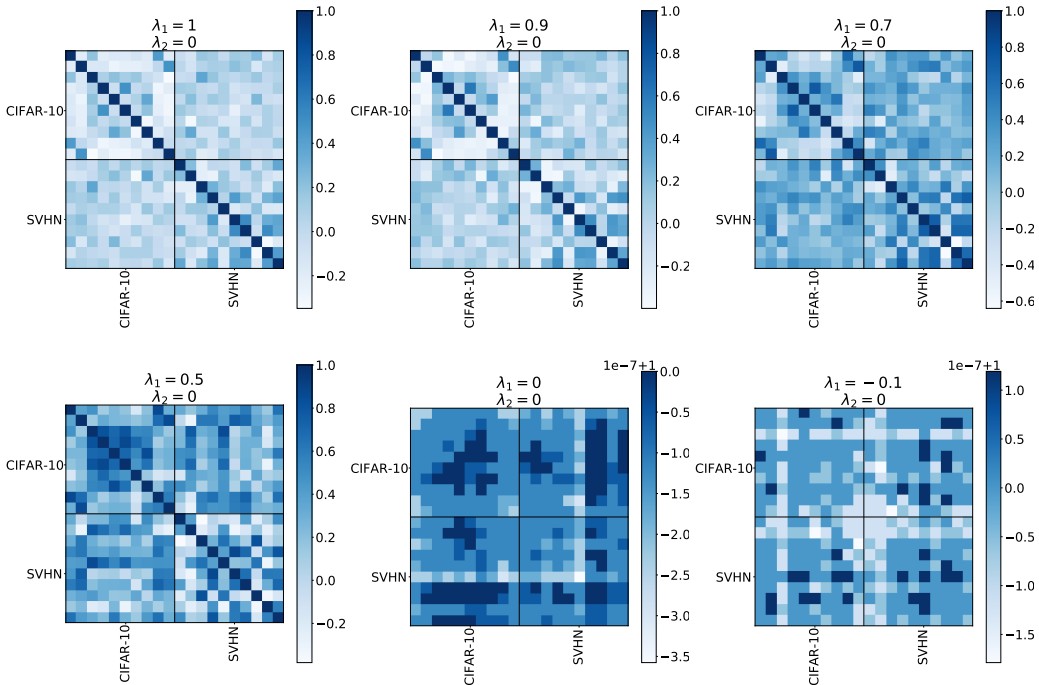

Figure A.1: **MDSSL trained on CIFAR-10 and SVHN with $\lambda_2 = 0$ and varying $\lambda_1$.** These heatmaps represent the similarity matrices between class-averaged representations of CIFAR-10 and SVHN. The first matrix in the first row represents our baseline, SimCLR (Chen et al., 2020). As $\lambda_1$ increases, the mutual similarity between samples within a domain increases. When $\lambda_1$ goes over 0 the mutual similarity between all training samples effectively reaches 1.

## A.4 RUNNING TIME OF MDSSL

The MDSSL optimization, as discussed in Section 3, is solved by iterating over the number of training datasets ($D$) in each term. Therefore, as the number of datasets increases, the running time of MDSSL will increase accordingly. We use a large batch size of 1024 which also accounts for increased running time for larger datasets. In Figure A.3, we see that as the number of training datasets increases, number of training hours of MDSSL also increases.

## A.5 IMPLEMENTATION DETAILS

Table 6 summarizes the entire architecture of each component of MDSSL with the filter and output dimensions for input image size $3 \times 32 \times 32$. Our implementation of MDSSL is on PyTorch. We

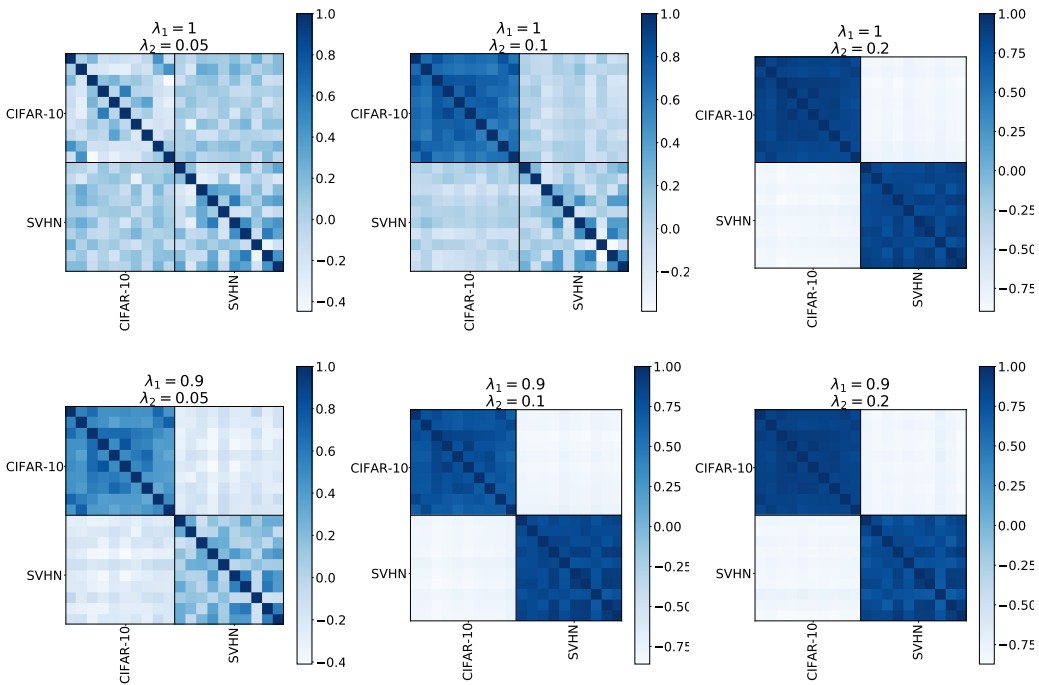

Figure A.2: **MDSSL trained on CIFAR-10 and SVHN with $\lambda_1 = -1$ (first row) and $\lambda_1 = -0.9$ (second row) and varying $\lambda_2$.** These heatmaps represent the similarity matrices between class-averaged representations of CIFAR-10 and SVHN. As $\lambda_2$ increases, the mutual similarity between samples across domains decreases. When $\lambda_1 = -0.9$ MDSSL learns to push samples within a domain closer while simultaneously reducing the similarity of samples across domains.

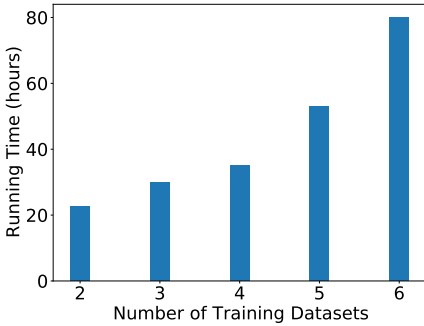

Figure A.3: **Running Time of MDSSL.** In this plot, we show how the running time increases when the number of training datasets increases in MDSSL. We train MDSSL on the following datasets in the given order: CIFAR-10, SVHN, STL-10, CIFAR-100, DTD, Tiny-ImageNet.

use ResNet-50 (He et al., 2016) as the base encoder for all our experiments. Since we have multiple datasets during training, we prepare a DataLoader for each dataset and load batches of size $1024$ from each dataset. We refer to these as *dataset batches*. When the number of training datasets is low, we concatenate all dataset batches ($\mathbf{X}$) and pass it through the encoder ($f(.)$) and projection head ($g(.)$) to get $\mathbf{Z}$. However, when the number of training datasets increases, $\mathbf{X}$ becomes too large and may require more memory to encode. In this case, we first separately encode every dataset batch and then concatenate all dataset embeddings to get $\mathbf{Z}$. This trick helps us efficiently train MDSSL on 2 GPUs with 4 training datasets and a high batch size of $1024$.

In Section 3.1, we discuss the experimental setup with hyperparameters for MDSSL training. We summarize these parameters in Table 7. We evaluate MDSSL using the linear evaluation protocol

Table 6: Architecture of MDSSL encoder, projection head and linear classifier

| MDSSL Component | Layer | Output Size | Filters |
|---|---|---|---|
| *ResNet-50 Encoder* | Conv2d | $64 \times 16 \times 16$ | $7 \times 7$, 64, stride 2, padding 3 |
| | BatchNorm | $64 \times 16 \times 16$ | 64 |
| | RelU | $64 \times 16 \times 16$ | - |
| | MaxPool2d | $64 \times 8 \times 8$ | $3 \times 3$, stride 2, padding 1 |
| | Bottleneck | $256 \times 8 \times 8$ | planes 64, blocks 3 |
| | Bottleneck | $512 \times 4 \times 4$ | planes 128, blocks 4 |
| | Bottleneck | $1024 \times 2 \times 2$ | planes 256, blocks 6 |
| | Bottleneck | $2048 \times 1 \times 1$ | planes 512, blocks 3 |
| | AdaptiveAvgPool2d | $2048 \times 1 \times 1$ | $1 \times 1$ |
| *Projection Head* | Linear | 2048 | 2048 |
| | RelU | 2048 | - |
| | Linear | 128 | 128 |
| *Linear Classifier* | Linear | 10 | 10 |

Table 7: Hyperparameter details for MDSSL encoder, projection head and linear classifier

| MDSSL Component | Parameter | Value |
|---|---|---|
| *Encoder and Projection Head* | Latent Dimension | 128 |
| | Temperature | 0.1 |
| | Optimizer | LARS |
| | LR Scheduler | Warmup-Anneal |
| | Learning Rate | 4 |
| | Weight Decay | $10^{-6}$ |
| | Batch Size | 1024 |
| | Number of Training Iterations | 48,000 |
| | GPU | Nvidia GeForce RTX 2080 |
| *Linear Classifier* | Input Dimension | 128 |
| | Optimizer | SGD |
| | LR Scheduler | - |
| | Learning Rate | 0.1 |
| | Weight Decay | - |
| | Batch Size | 1024 |
| | Number of Training Iterations | 30000 |
| | GPU | Nvidia GeForce RTX 2080 |

(Kolesnikov et al., 2019; Bachman et al., 2019; van den Oord et al., 2019). At test time, we discard the projection head ($g(.)$) and keep only the ResNet encoder ($f(.)$). We freeze the encoder and define a trainable linear layer that maps 128-dimensional features from the encoder to class probabilities. This is our linear classifier. We train this classifier over the frozen embeddings from the ResNet encoder for 100 epochs with a batch size of 1024. We use the SGD optimizer with an initial learning rate of 0.1. We summarize all of these parameters in Table 7. We optimize the linear classifier using the cross-entropy loss and calculate the top-1 accuracy at the end of training.

