# OpenReview forum: "Multi-Domain Self-Supervised Learning"
_ICLR.cc/2022/Conference — ICLR 2022 Submitted_

### Official Review · Reviewer_ZUR1 · 2021-10-25

**Correctness:** 3
**Technical Novelty And Significance:** 2
**Empirical Novelty And Significance:** 3
**Recommendation:** 5
**Confidence:** 4

**Main Review:**

Ups

1.	Simple solution. The learning objective includes 3 terms, the first 2 terms (minimize the discrepancy between data with different views and maximizing the data pair from the same domain) are derived from SimCLR and the 3rd term is designed to maximizing the discrepancy between pair of data from different domains.
2.	Good results. 25% increase in top-1 accuracy when comparing to SimCLR trained on the union of multi-domain data.
3.	Great presentation and through experimental study. The table presented in the paper are well organized and informative to present every details. Clear and concise.

Downs:
1.	It is not very clear for the motivation in Eq (3). Since \lambda_1 is always negative, does it mean the approach is trying to maximize the discrepancy of a pair of data from the same domain and maximize those between a pair of data from different domain even further? The choice of \lambda_1 and \lambda_2 seems very sensitive, could the author provide the corresponding set-up for each experiment and list them in each table?
2.	It is unclear how one domain data is benefits from others. SimCLR performs pool on unobserved domains. This seems that multi-domain data are crossed in the latent space. MDSSL is superior because it introduces a large margin between each domain data. It hints that performance of a particular domain is upper bounded by SimCLR on the single domain. Is it as expected? How will other domains data help?
3.	Within domain clusters seems important and may need further research. Since MDSSL performs better than SimCLR on the union of multi-domain data, does MDSSL perform superior on single domain with each cluster as a new domain?


**Summary Of The Paper:**

The paper extends self-supervised contrastive learning from one single domain to multiple domains and achieve better generation for multi-domain performance.

Intuitively, the paper finds SimCLR, an existing self-supervised contrastive learning algorithm performs pool when takes the union of a multi-domain data (e.g., union of CIFAR10, STL-10, SVHN etc) , and is also weak when training on one specific domain and evaluates on the unobserved domain. To bridge the gap, the paper brings in an additional loss into SimCLR to mitigate the gap across different domains - maximizing the discrepancy between data from different domains.

This change achieves great improvement when training and evaluating via a unified model and better generation – train on one domain data and evaluate on the other domains.


**Summary Of The Review:**

Pros: Simple approach and good results for building a unified model for multi-domain data.

Cons: The formulation and motivation may need further clarification.

---

> ### Author Response · Authors · 2021-11-18
> **Expression (3), multi-domain motivation and clustering**
>
> We thank reviewer ZUR1 for constructive comments and feedback. We address the questions raised below.
>
> 1. Clarifying the motivation for expression (3): In our previous draft, we specified $+\lambda_1$ in Term 3 but we only used negative values for $\lambda_1$. So, similar to SimCLR, we also aim to minimize the similarity between samples within a domain. But we use $\lambda_1$ to control the extent to which samples should be dissimilar. We observe empirically, that using a value slightly less than 1 for $\lambda_1$ produces better latent structure as shown in Appendix Section A.2. To reduce this ambiguity, we have modified our optimization to have $- \lambda_1$, where now $\lambda_1$ is a positive regularization parameter.
>
>    We have also added the corresponding values for $\lambda_1$ and $\lambda_2$ for each experiment in Table 1 and Table 3. We observe that fixing $\lambda_1 = 0.9$ and $\lambda_2 = 0.1$ generalizes well across all experiments and outperforms the performance of SimCLR. We mildly tune the regularizers to boost the performance further.
>
>
> 2. How one domain data benefits from others: MDSSL is superior to SimCLR because we train a single encoder over multiple domains and also ensure that the domains are distinguishable in the latent space. The main benefit of training over multiple domains is the generalization power of the encoder as shown in Table 3. We observe that more diverse training domains help MDSSL generalize better. In Figure 2, we observe that increasing the number of domains consistently increases the performance of each constituent domain. In most of our experiments, the performance of MDSSL is upper bounded by SimCLR on a single-domain. However, in Figure 2, we observe that if we have a high number of diverse training domains, we can achieve a modest improvement over SimCLR on CIFAR-10.
>
>
> 3. Within domain clusters: MDSSL with clustering performs well, especially on domains that are more visually separable. In our experiments, clustering is done across different domains. Clustering cannot be done on a single-domain setup because we would have to know the number of classes in that domain (the information we are not assuming to have). Unlike clustering across domains, clustering within a domain is much more unstable because classes can overlap heavily in the latent space. Therefore, we only apply MDSSL with clustering to multi-domain setups.

---

> > ### Comment · Area_Chair_axAS · 2021-12-01
> > **request feedback on authors' reply**
> >
> > Hi Reviewer ZUR1,
> >
> > Thank you for your reviews. Does the authors' reply answer your questions and concerns? what is your feedback? Thanks.

---

> > > ### Comment · Reviewer_ZUR1 · 2021-12-02
> > > **Response to rebuttal**
> > >
> > > Thank the author for the response.
> > >
> > > For Q1, please make sure to make it clear because the intuition and meaning seems kind of confused in current version.
> > >
> > > For Q2, I cannot totally agree. It is reasonable to distinguish multi-domain data in the latent space to improve the predictive power. However,  only enforce large margin between multi-domain may not be strong enough. When considering multi-domain data with large overlaps, there could be a strong trade-off, when the parameter is small, the model searching for alignment, otherwise search for brand new decision boundary. So \lambda should be sensitive. I don't observe enough intuition and new idea here. Thus, even though the performance is encouraging, I think the significance seems incremental.
> > >
> > > In summary, the idea is simple and the performance is encouraging for multi-domain set-up. However, the significance of the idea and the motivation can be further improved. This is borderline work to me. So I prefer to maintain my rating "5: marginally below the acceptance threshold"

---

> > > > ### Author Response · Authors · 2021-12-02
> > > > **Author Response**
> > > >
> > > > We thank Reviewer ZUR1 for their response and address each comment below.
> > > > 1. We will ensure that we explain the intuition behind the two regularizers more clearly in our final draft.
> > > >
> > > > 2. We believe the reviewer is not correct here: Since the $\lambda_2$ term is always minimized in our method, the model always searches for a decision boundary between domains, regardless of the value of $\lambda_2$. The model never searches for an alignment as long as $\lambda_2 > 0$. We agree that in cases where multi-domain data has large overlaps, $\lambda_2$ should be tuned carefully to ensure that distinguishability is maintained. This can be empirically illustrated in our visualization of similarity matrices as shown in Figure A.2 in our Appendix. We will add more explanations about this in the final draft.
> > > >
> > > > 3. We are a bit confused with this comment saying "the performance is encouraging, I think the significance seems incremental." We are glad the reviewer has found the performance of our method encouraging. However, to get such a performance on several benchmark datasets, our method involves several novel and non-trivial components: (i) We have introduced a three-level self-supervised loss function which is essential to train over multi-domain data, without which models only achieve sub-par performance, (ii) Our MDSSL loss is also necessary to help models generalize better to unseen domains, (iii) We also make it possible to train MDSSL with clustering in a robust fashion when domain labels are unknown.

---

> > > > > ### Comment · Reviewer_ZUR1 · 2021-12-05
> > > > > **Review Response**
> > > > >
> > > > > 2. I understand the model is always searching for a decision boundary between domains. My point is for multi-domain data with larger overlaps, e.g., some classes are exactly the same and some are not. The model would searching for alignment based solutions (\lambda_2 < 0), align the domain data with exact the same classes and make clear decision boundary for distinct classes. For multiple-domain data without class overlaps, \lambda_2 > 0 should make sense. However, the paper lacks such analytics and the trade-off study is not included (this can be done using simulation data), where the case \lambda_2 = 0 is. This part raise my concern when extending the model to different multi-domain data. Is the multiple domain data used in the paper cherry picked? What if we use multiple domain data with overlaps, e.g., ImageNet 1K and ImageNet, or multiple domain data without overlaps ImageNet 1K and  satellite dataset.
> > > > >
> > > > > 3. My point is performance-wise the result is encouraging. Idea-wise the significance seems incremental. The key idea is introducing a regularization term in Eq (4), which minimize the agreement between samples from different domains. This is quite simple and I am afraid this is the task specific simple set-up, the value can be kind of limited to the majority of community.
> > > > >
> > > > >  This is a borderline work to me, so I keep my original rating. There are contributions (a simple multi-domain self-supervised learning solution and good results), and flaws (in-depth analytics and ablation studies).

---

> > > > > > ### Author Response · Authors · 2021-12-06
> > > > > > **Author Response**
> > > > > >
> > > > > > We thank Reviewer ZUR1 for their response.
> > > > > >
> > > > > > 2. We would like to highlight Table1, Table 3 and Figure 5 where we provide the example of multi-domain setups including CIFAR-10 and STL-10 which overlap on 9 out of 10 classes. We believe this is an appropriate example for largely overlapped multi-domain setups that the Reviewer describes. Our observation on such setups is that, with appropriate selection of $\lambda_2 > 0$, the model does reasonably well in separating both datasets. In the camera ready version, we will ensure to add further analysis comparing both setups (overlapping and non-overlapping) as requested by the reviewer.

---

### Official Review · Reviewer_KFWz · 2021-11-02

**Correctness:** 3
**Technical Novelty And Significance:** 2
**Empirical Novelty And Significance:** 2
**Recommendation:** 6
**Confidence:** 4

**Main Review:**


Strengths:

The method is clear and clean, and the paper is well organized and laid out. The results also show some nice improvements when working with multiple datasets. Overall, it is a nicely written paper with a clear contribution.

Weaknesses:

There are some shortcomings in the paper, and some spots where the language should be revised.

First, there is a confound of dataset size that the authors should address. The authors focus on very small datasets (SVHN, CIFAR10, and similar), then show that SimCLR does poorly to generalize when you mix them all together. What about if working with very large diverse image datasets, like the full size ImageNet? This should at least be mentioned, since these SSL techniques are typically intended for ImageNet and larger training.

Second, there is not a good theoretical justification for expression (3). The heart of the method is the contrastive learning objective of the expressions (2-4).  (2) and (3) give you the standard SimCLR objective, which puts same-images close together in the latent space, and different-images farther apart in the latent space.

But there is a small tweak done here. The authors essentially take expression (2), and break it into 2 cases, the case of 2 different images within the database (they now make the representations CLOSER instead of farther like simCLR), and the case of 2 different databases (expression 4), where they make the representations farther apart (that makes sense). I do not have a theoretical appreciation for the tweak of expression (3), to make different images within a database have similar representations, and why that is what is making the representations generalize better. Is it doing some kind of regularization? Can the authors provide some more justification for this change, and perhaps give a bit more of the SimCLR context as I have described here?


minor:


- related work, 2nd paragraph, first sentence does a bad job of explaining the goal of SSL. Rephrase to say something like “SSL learns representations that are blah blah. It achieves this by … (then the thing you said). Right now, it doesn’t do SSL justice.


**Summary Of The Paper:**


The paper presents a contrastive learning approach to self-supervised learning (SSL) that works for multiple domains.

They show that their approach generalizes better across domains than the baseline algorithm (SimCLR). Their method is basically an extension to SimCLR, which is nice because then SimCLR acts as the perfect baseline model.

They also provide an approach to using multiple domains when the domain labels are not available (using clustering).


**Summary Of The Review:**


The paper is clean and shows some improvements, but there is not a clear theoretical argument. Furthermore, the paper is limited by using tiny datasets which limit how well we would expect these results to generalize in practice.

---

> ### Author Response · Authors · 2021-11-18
> **ImageNet experiments, explaining expression (3)**
>
> We thank reviewer KFWz for constructive comments and feedback. We address the questions raised below.
>
> 1. ImageNet experiments: SimCLR requires large amount of memory and compute which makes it expensive to train on datasets like full-sized ImageNet under limited computational resources. Due to these resource constraints, we use 32x32 sized images in all of our experiments. Therefore, it is unfair to compare our results with a SimCLR model pre-trained on 224x224 samples of ImageNet. Nevertheless, we have still added these results in our updated draft in Section A.2. We observe that MDSSL outperforms SimCLR pre-trained on full ImageNet in terms of generalization.
>
>     In our main experiments, we instead use Tiny-ImageNet dataset as it is significantly larger and more diverse compared to the remaining datasets. In Table 3, we observe that simply using a large and diverse dataset like Tiny ImageNet does not suffice to have generalizable embeddings since it generalizes poorly on visually dissimilar domains like SVHN. However, training MDSSL on multiple domains, including Tiny ImageNet, helps the model generalize better even on datasets like SVHN.
>
>     Given our limited resource and time constraints, we were able to add some new experiments on a subset of ImageNet (with 250K samples) which includes 1000 classes, with each class having 250 samples, all resized to 32x32 in Table 3 in our updated draft. We observe that single-domain and multi-domain SimCLR do not generalize well to unseen domains especially domains like SVHN. MDSSL on ImageNet (250K), CIFAR-100 and SVHN improves the generalization capability. This further validates our findings that multi-domain training using MDSSL is beneficial even for highly diverse domains like ImageNet. We will ensure that we have full ImageNet (32x32) results ready in our final draft.
>
>
> 2. Justification for expression (3): Term 3 in our optimization problem controls the agreement between samples within a domain. In our previous draft, we specified $+\lambda_1$ in Term 3 but we only used negative values for $\lambda_1$. We realized this could be confusing and in the updated draft, to reduce this ambiguity. We have modified our optimization problem to have $- \lambda_1$ in the objective, where now $\lambda_1$ is a positive regularization parameter. Therefore, similar to SimCLR, we also aim to minimize the similarity between samples within a domain. We use $\lambda_1$ to control the extent to which samples should be dissimilar in each domain. We observe empirically, that using a value slightly less than 1 for $\lambda_1$ produces improved latent structure as shown in Appendix Section A.2.
>
>
> 3. Explaining the goal of SSL: Thank you for this suggestion on changing the language around self-supervised learning in the Related Work section. We have incorporated this change in our updated draft.

---

> > ### Comment · Reviewer_KFWz · 2021-11-29
> > **rebuttal response**
> >
> > Thank you for addressing the main concerns and adding some experiments to show results with more varied data.
> >
> > I think adding the regularization to control how different within-domain examples look still lacks some theoretical rigor, but that could be a place for future work. I am convinced from the experiments that it does have the intended effect.

---

### Official Review · Reviewer_stxw · 2021-11-07

**Correctness:** 4
**Technical Novelty And Significance:** 2
**Empirical Novelty And Significance:** 3
**Recommendation:** 6
**Confidence:** 4

**Main Review:**

Strengths:
1. The paper is clearly written and well-organized: The motivation, method, experiments, and results are clearly explained. The loss formulation for MDSSL is sensible.
2. All experiments show substantial improvement in performance on downstream tasks over single-domain SSL and naive combination of domains.
3. Ablation studies on hyperparameters and t-SNE plots are helpful in understanding the effect of  design choices on performance.

Weaknesses (Not weaknesses per se, but some questions/suggestions):
1. No external baselines: As the authors mention in the paper, the Feng et al. approach based on mutual information is a theoretically well-motivated approach for combining multiple domains for learning SSL representations. Comparing MDSSL with this baseline will strengthen the paper.
2. Additional analysis along domain diversity: While the authors use fairly diverse datasets in terms of size and label types, they have only used natural, object-focused image datasets. Additional experiments with other data domains---medical images, satellite images etc.---will strengthen the paper.
3. The paper is missing an experiment with full ImageNet pretraining (which is common in SSL papers), which could clarify if MDSSL has advantage over learning representations using a single, but large and diverse dataset.


**Summary Of The Paper:**

This paper proposes a method for multi-domain self-supervised representation learning (MDSSL).  The contrastive MDSSL training objective consists of three  terms: (1) a loss  term  that  maximizes the similarity between two transformed views of a sample in the latent space (identical to SimCLR), (2) a loss term that maximizes the similarity between every pair of samples within a dataset,  and  (3)  a loss term  that minimizes the similarity between pairs of samples  across different datasets. Loss terms (2) and (3) are weighted using hyperparameters. Authors perform experiments using common datasets including CIFAR-10, CIFAR-100,  STL-10, TinyImageNet and  SVHN. Experiments show that MDSSL outperforms (1) representations learnt by naively combining all datasets and (2) representations learnt on individual domains,  on average classification accuracy on training domains as well as held-out domains. Authors also perform  ablation studies to explore the effect of loss term hyperparameters on  downstream performance. The paper further extends the basic MDSSL method by adding an assumption that dataset labels are not available, and determine dataset labels using k-means clustering during the training process, and also explore a robust k-means clustering method. Experiments using 2 training domains show that the downstream classification performance of this unsupervised MDSSL method is comparable to original labeled MDSSL.

**Summary Of The Review:**

The paper proposes a simple and sensible method for  multi-domain self-supervised learning that obtains good performance on for both in-domain and out-of-domain problems. Comparison with baselines and experiments with full ImageNet datasets is missing. Overall, I think the paper is marginally above the acceptance threshold.

---

> ### Author Response · Authors · 2021-11-18
> **External baselines, domain diversity and ImageNet experiments.**
>
> We thank Reviewer stxw for constructive comments and feedback. We address the questions raised below.
>
> 1. External baselines: The work by Feng et al. is a relevant baseline to compare. However, their implementation is not available to test on our multi-domain setup. We have requested the authors for their code. If we gain access, we will include the baselines in our final draft.
>
>
> 2. Additional analysis along domain diversity: Thank you for this suggestion. We have added new set of experiments in the Appendix of our revised draft, containing image datasets that are not object-focused. Specifically, we included the datasets EuroSAT and Chest X-Ray (Pneumonia), and combined these with DTD which is also non-object-focused. Please refer to Section A.1 in our updated draft. Similar to the object-focused datasets, we observe that the single-domain setup do not generalize well. MDSSL on these datasets performs consistently well on each domain and almost matches the single-domain baselines.
>
>
> 3. ImageNet experiments: SimCLR requires large amount of memory and compute which makes it expensive to train on datasets like full-sized ImageNet under limited computational resources. Due to these resource constraints, we use 32x32 sized images in all of our experiments. Therefore, it may be unfair to compare our results with a SimCLR model pre-trained on 224x224 samples of ImageNet. Nevertheless, we have still added these results in our updated draft in Section A.2. We observe that MDSSL outperforms SimCLR pre-trained on full ImageNet in terms of generalization.
>
>    In our main experiments, we instead use Tiny-ImageNet dataset as it is significantly larger and more diverse compared to the remaining datasets. In Table 3, we observe that simply using a large and diverse dataset like Tiny ImageNet does not suffice to have generalizable embeddings since it generalizes poorly on visually dissimilar domains like SVHN. However, training MDSSL on multiple domains, including Tiny ImageNet, helps the model generalize better even on datasets like SVHN.
>
>    Given our limited resource and time constraints, we were able to add some new experiments on a subset of ImageNet (with 250K samples) which includes 1000 classes, with each class having 250 samples, all resized to 32x32 in Table 3 in our updated draft. We observe that single-domain and multi-domain SimCLR do not generalize well to unseen domains especially domains like SVHN. MDSSL on ImageNet (250K), CIFAR-100 and SVHN improves the generalization capability. This further validates our findings that multi-domain training using MDSSL is beneficial even for highly diverse domains like ImageNet. We will ensure that we have full ImageNet (32x32) results ready in our final draft.

---

> > ### Comment · Reviewer_stxw · 2021-12-01
> > **Updated response**
> >
> > Thanks for your thoughtful response. I am happy with the additional experiments and I will keep my recommendation for acceptance.

---

### Author Response · Authors · 2021-11-19
**Thank you for your valuable feedback**

We would like to thank all the reviewers for taking your time in reviewing our work and providing valuable feedback. In our responses, we did our best to address all of your comments. Since the discussion period is closing soon, we would appreciate it if you let us know if you have any further comments or questions. Thank you.

---

### Decision · Program_Chairs · 2022-01-20

**Decision:**

Reject

**Comment:**

This paper introduces a multi-domain self-supervised representation learning method. Its objective consists of three terms: the first two terms are identical to SimCLR and the last one is to minimize the similarity of pairs across different datasets which is similar to the second term of SimCLR. In the experiment, it tests the methods across multiple common datasets. The method is simple but results are pretty good at the multi-domain setting. It seems to demonstrate the importance of domain clustering and moving the domains apart. However, there are several important questions the paper may need more clarification on:
1. What is the definition of the domain? How to determine the pair of data is from different domains? What is the motivation/theory that you used to choose those datasets as different domains in your experiment?
2. Is there any of the public datasets that would cover multiple domains?
Without solving these questions, I think it would constrain the future research/adoption of the method.